# Autocompletion of Code from Keywords

## Abstract

We present a simple approach for synthesizing code from keywords. Our system takes *keywords*, a subset of tokens from the target code, and generates a line of code. The main challenge is that we only have the output of the system and do not know the real input distribution. We construct simple synthetic distributions by dropping each keyword randomly and based on its frequency, which still generalize to various test distributions. We train a standard sequence-to-sequence model on the synthetic training data and successfully synthesize 71.5% of examples in our manually generated test set. Our approach is simple and language-agnostic and therefore can be easily extended and applied to any programming language.

## 1. Introduction

Developers often have to repeatedly write boilerplate code, or they may not remember the exact syntax of the desired code. Most IDEs try to alleviate this problem by autocompleting one token at a time.[1] However, they require users to know and specify all tokens in the target code in a left-to-right manner to get suggestions. Other tools such as snippets[2,3] generate multiple tokens at once by simply replacing a string with another string, e.g., `if main` with `if __name__ == "__main__":`, based on hard-coded mappings. In this case, users have to remember and maintain all the mappings.

In this paper, we present a new type of autocomplete system to help developers become more efficient in an actual programming environment. We define our task as follows: given a set of *keywords*, i.e., a subset of tokens from desired

| Input | Output |
|---|---|
| `for i 10` | `for i in range(10):` |
| `import np` | `import numpy as np` |
| `class example` | `class Example():` |
| `except e` | `except Exception as e:` |
| `with tf sess` | `with tf.Session() as sess:` |

*Table 1.* Example inputs and outputs synthesized by our system

code, we want to synthesize a line of code. Our method mainly targets boilerplate code as well as common structures and patterns in source code. We target Python in this work, but our approach does not require any language-specific preprocessing and therefore can be easily applied to any programming language. Table 1 shows example inputs and corresponding outputs produced by our system. More examples are listed in Appendix A.

Despite having a vast amount of available source code, which serves as the output of our system, we do not know what the user input will be like. In other words, we need to model how terse or verbose users will be, or which tokens they will keep or drop. To this end, we create artificial training data from synthetic distributions and use a standard sequence-to-sequence model (Sutskever et al., 2014) with attention (Luong et al., 2015) and copy mechanisms (Gu et al., 2016) to synthesize code from keywords.

We evaluate our system with two corpora: one is manually constructed to capture the gist of our system's ability, and the other is generated from a GitHub repository to gain a sense of how much benefit users can get in a real-world programming environment. Our method can synthesize 71.5% of common use cases, generating 10.24 tokens from 5.28 keywords on average.

## 2. Problem Statement

Our system takes a sequence of keywords $x = (x_1, ..., x_n)$ as input and produces a sequence of tokens $y = (y_1, ..., y_m)$ as output. Here, we consider the case where $x \subseteq y$, i.e., the keywords are a subset of the output tokens.

Note that keywords are a *subset* of the output tokens, not a subsequence, as we do not require the order of tokens to be preserved, and any combination of (not necessarily consecutive) tokens from the output can be selected.

---

[1]Anonymous Institution, Anonymous City, Anonymous Region, Anonymous Country. Correspondence to: Anonymous Author <anon.email@domain.com>.

Preliminary work. Under review by the International Conference on Machine Learning (ICML). Do not distribute.

[1]https://code.visualstudio.com/docs/editor/intellisense
[2]https://github.com/honza/vim-snippets
[3]https://github.com/SirVer/ultisnips

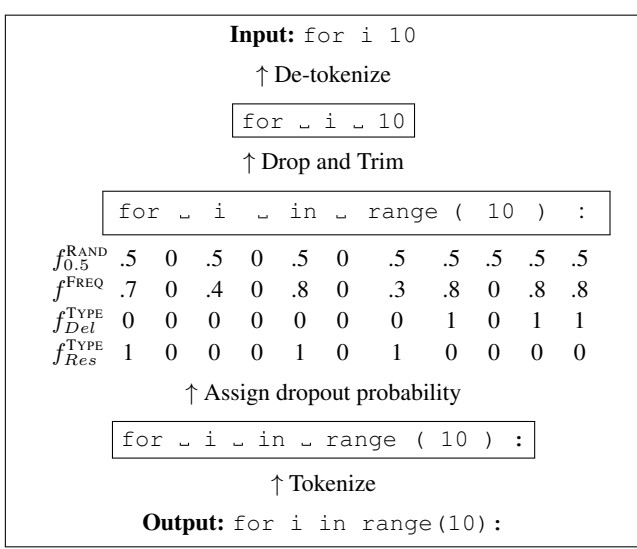

| | for | ␣ | i | ␣ | in | ␣ | range | ( | 10 | ) | : |
|---|---|---|---|---|---|---|---|---|---|---|---|
| $f^{\text{RAND}}_{0.5}$ | .5 | 0 | .5 | 0 | .5 | 0 | .5 | .5 | .5 | .5 | .5 |
| $f^{\text{FREQ}}$ | .7 | 0 | .4 | 0 | .8 | 0 | .3 | .8 | 0 | .8 | .8 |
| $f^{\text{TYPE}}_{Del}$ | 0 | 0 | 0 | 0 | 0 | 0 | 0 | 1 | 0 | 1 | 1 |
| $f^{\text{TYPE}}_{Res}$ | 1 | 0 | 0 | 0 | 1 | 0 | 1 | 0 | 0 | 0 | 0 |

*Table 2.* Overall process of generating input and output pairs. The dropout probability for each token is sampled from the random, frequency-based, and type-based distributions, respectively.

## 3. Synthesizing Datasets

In this section, we describe our main contribution of creating training data from synthetic distributions, which can generalize to various real test distributions. Table 2 shows how we derive a sequence of keywords (input) from a line of original code (output), producing an input-output pair. For each line of code, we first tokenize, assign a probability of being dropped to each token, drop tokens based on the probability, trim by inserting and removing whitespace, and de-tokenize the result to generate input keywords. We repeat this process line by line and use the input-output pairs as our training data. For test sets, keywords are derived in the same way, but we utilize the prior knowledge of the target language to create tailored test sets.

**Step 1. Abstraction** Before tokenization, we generate two output sequences from each line of code, once with and once without the arguments, e.g., `for i in range(10)` and `for i in range()`. For simplicity, we consider any token between a pair of matching parentheses to be an argument and remove all arguments without considering several or nested functions separately. It can be seen as abstraction in that we discard case-specific arguments and focus on general structures of code. By doing so, our model can learn when to synthesize general structures as opposed to always trying to synthesize concrete code.

**Step 2. Tokenization** We use a standard tokenization based on non-alphanumeric characters, underscores, and camel cases. Additionally, we use four special tokens to (1) distinguish string literals, (2) distinguish number literals, (3)

capitalize the first letter of the next token, and (4) capitalize all letters in the next token. In order to identify strings, we restrict strings to be placed between quotation marks. De-tokenization is the reverse of tokenization.

**Step 3. Dropout probability** We propose several ways to model the input distribution. During training, we do not require any prior knowledge of the target language to model the input distribution. Instead, we assign a probability of being dropped to each token based on random and frequency-based distributions.

Given a sequence of keywords $x = (x_1, ..., x_n)$ as input, we want to compute the probability of each token $x_i \in x$ being dropped. As a baseline, we drop every token with a fixed probability of $p$ as follows:

$$f^{\text{RAND}}_p(x_i) = p.$$

Alternatively, we can drop tokens based on their frequency to keep rare tokens such as variable names and literals as follows:

$$f^{\text{FREQ}}(x_i) = \frac{\text{freq}(x_i)}{\max_{v \in V} \text{freq}(v)}$$

where the frequency of each token is normalized by the highest frequency of a token in the entire vocabulary. In practice, we scale the probability to be in the range between $0$ and $0.8$.

In testing time, we utilize the prior knowledge of the target language to drop *all* tokens with specific properties and evaluate our system's ability on recovering certain types of tokens. Here, we consider two token types, delimiters and reserved words, since other types such as operators, identifiers, and literals are almost impossible to be recovered when dropped. We identify and drop tokens of the type $t$ as follows:

$$f^{\text{TYPE}}_t(x_i) = \begin{cases} 1 & \text{if type}\,(x_i) = t \\ 0 & \text{otherwise.} \end{cases}$$

For instance, if we want to drop reserved words from the example in Table 2, we assign 1 to `for`, `in`, and `range` and 0 to the other tokens.

**Step 4. Trimming** To resemble a realistic user input query, we insert and remove whitespace. While dropping tokens, we keep all the whitespace and replace each dropped token with a single whitespace. Then, we remove redundant whitespace at the beginning and end of the sequence and aggregate any consecutive whitespace between tokens. For example, if the result of drop is ␣ i ␣ ␣ 10, we trim starting and trailing whitespace and aggregate consecutive whitespace to get the final input keywords i ␣ 10.

## 4. Model

We use a standard sequence-to-sequence model (Sutskever et al., 2014) with attention (Luong et al., 2015) and copy mechanisms (Gu et al., 2016). We first encode input keywords using the encoder and use the decoder to generate tokens from a fixed vocabulary or copy tokens from the input. The use of attention helps the model learn long-distance dependencies over input and output tokens, and the copy mechanism allows the model to output unseen tokens such as variable names by copying from input keywords.

## 5. Experiments

In this section, we describe the setup for our experiments, report results, and analyze errors. Our main findings are as follows: the frequency-based distribution achieves considerably high accuracy on the manually generated test set, demonstrating the effectiveness of our system on its main use cases. The baseline random dropout distribution performs well, even comparable to the frequency-based distribution. Both of them result in relatively low accuracy on automatically generated test sets, because the input of these test sets often fails to retain all necessary keywords to recover the original code.

**Data collection.** For training, we use over 800 GitHub repositories containing 200K files and 2M lines of code. We remove empty lines and comments and sample twice from each line, once with and once without arguments. Then, we select lines that have at least one alphanumeric token and occur more than 10 times to rule out irregular lines, resulting in about 1.5M lines of code.

For testing, we use two corpora. First, we use carefully constructed 200 input and output pairs to represent the real user input distribution. This manual test set is designed to reflect simple, common code usage, as our system aims to learn general patterns and structures. Second, we use a single GitHub repository, which contains 3K lines of code, to understand how much benefit users can get in a real-world programming environment. We observe that the manually constructed keywords tend to include not only rare tokens like case-specific arguments, but also common tokens such as def to specify the intention. The pairs used in the manual test set are listed in Appendix A and B.

**Input distributions.** We create training data by sampling from the random and frequency-based distributions, while varying the value for the parameter $p$. We name a trained model with the name of the distribution used to generate its training data, e.g., RAND 0.5 for the model trained with the data generated with $f_{0.5}^{\text{RAND}}$.

For test sets, we construct one test set from the manually

|  | Manual |
|---|---|
| RAND 0.1 | 32.5 |
| RAND 0.3 | 49.5 |
| RAND 0.5 | **60.0** |
| RAND 0.7 | 58.0 |
| RAND 0.9 | 42.0 |
| FREQ | **71.5** |

Table 3. Accuracy of models trained on the random and frequency-based distributions and evaluated on the manual test set

| Rank | Accuracy |
|---|---|
| 1 | 71.5 |
| 2 | 76.5 |
| 3 | **78.0** |

Table 4. Accumulative accuracy for the top 3 suggestions of FREQ

generated corpus by selecting keywords by hand, and four test sets from a repository by automatically deriving input keywords using $f_{0.5}^{\text{RAND}}$, $f^{\text{FREQ}}$, $f_{Del}^{\text{TYPE}}$, and $f_{Res}^{\text{TYPE}}$. We name a test set with the name of the distribution, e.g., Manual for the manually generated test set and Delimiters for the automatically generated test set with $f_{Del}^{\text{TYPE}}$.

**Implementation.** Our model consists of a two layer bidirectional LSTM encoder and a unidirectional decoder with 512 hidden units. We use stochastic gradient descent with a learning rate starting from 1.0 with decaying by 0.5 and a dropout of 0.3 at the input of each state. We jointly learn the 512-dimensional word embedding while training.

**Accuracy.** To measure the performance of our model, we compare the output of the system with the target code and consider the synthesis to be successful only if the two strings are identical. Note that this type of exact match is the most conservative way of evaluating the result for three reasons:

1. There can be multiple reasonable outputs for a single input. For instance, if `none` is the input, both `if None:` and `return None` are reasonable outputs.

2. The functional equivalence of programs is undecidable. The exact match between `class A:` and `class A():` fails, although they are semantically equivalent.

3. Even though the system generates nearly correct output, it can still struggle with synthesizing parts of the output such as variable names.

Comparing with top $n$ suggestions of the system or evaluating based on edit distance could mitigate this problem. In this work, we report all results based on exact match with the top 1 suggestion by default.

| Automatically Generated Test Sets | | | | |
|---|---|---|---|---|
| | Rand | Freq | Deli | Rese |
| RAND 0.5 | **5.53** | 17.28 | 14.74 | 34.09 |
| FREQ | 3.81 | **18.40** | **18.97** | **39.07** |

*Table 5.* Accuracy of the two best distributions on automatically generated test sets. Both of them achieve relatively low accuracy, because the automatically generated test sets often fail to retain necessary keywords to recover the original code.

### 5.1. Evaluation on Manually Generated Test Set

Table 3 shows the performance of the models trained with different distributions and evaluated on the manually-generated test set. We find that the model trained on the frequency-based distribution FREQ yields the best performance of 71.5%, while the best model with the random distribution RAND 0.5 results in the comparable accuracy of 60%. We observe that when tokens are dropped too aggressively (RAND 0.9) or conservatively (RAND 0.1), the performance can degrade significantly.

As shown in Table 4, our best model FREQ synthesizes 78% of pairs within the top 3 suggestions based on beam search and ranks the correct code as the top 1 suggestion 91.67% of the time. It generates 10.24 tokens from 5.28 input keywords on average.

We categorize common errors made by the system and list examples of errors in each category in Appendix B. We observe that even when the system fails to synthesize the desired code, it tends to generate (seemingly) syntactically correct output. However, it often fails to copy all input keywords, format syntax, find proper tokens to insert, or keep irrelevant tokens from being inserted.

### 5.2. Evaluation on Automatically Generated Test Sets

Table 5 shows the performance of the two best models, RAND 0.5 and FREQ. Overall, the accuracy of both models is much lower than that on the manually generated test set. They both achieve low accuracy of 5.53% on the random test set, moderate accuracy of 18.97% on the frequency and delimiters test sets, and relatively high accuracy of 39.07% on the reserved words test set, mainly because of the easy cases such as `import x`, `def x`, and `class x`.

One of the main reasons for such low accuracy on the auto-generated sets, especially the random and frequency test sets, is that the derived input keywords often fail to retain all tokens necessary to fully recover the original code. For example, it is impossible to recover `if not x:` when either `not` or `x` is dropped.

We also notice that even when only certain type of tokens are dropped, the model has to reason about all possible types of tokens and often outputs the most probable code.

Consider the case `result`. Although the model generates the reasonable output `return result`, the original code `result = []` contains case-specific delimiters, which is hard for the model to predict solely from the given input.

Lastly, these test sets contain many rare tokens, e.g., user-defined variables, as well as repetitions of nearly-identical lines, e.g., adding arguments to a parser. This increases the difficulty of synthesis and can overly penalize a model for failing a set of similar inputs.

## 6. Related Work and Discussion

We strike a balance between existing autocomplete tools and recent data-driven approaches to code generation. With most existing tools, it is extremely difficult to find the desired code when users do not know tokens in the middle, since the tokens must be specified in sequential order. Moreover, even if users know every token in the target code, they still need to type or find roughly the same number of tokens one by one. In contrast, our system allows users to specify a subset of tokens to get the complete line.

Recently, many data-driven approaches have been proposed to generate or retrieve the code from natural language descriptions (Yin & Neubig, 2017) or natural language-like labels (Murali et al., 2018). Their output tends to be richer, spanning one to multiple lines of code, in order to synthesize code that performs the task specified by users. However, they serve more as tailored search engines rather than built-in tools for a programming environment. On the other hand, our system can be easily incorporated with the existing environment and used to write each and every line of code.

By being language-agnostic, however, there are limitations in our approach. Since our method is not aware of the context of input, it often fails to generate user-defined tokens such as variable names, whereas they are easily inferred by most existing tools. Another major limitation is that it does not guarantee the syntactic correctness of the output. Most previous work heavily relies on the prior knowledge of syntax and semantics to represent or reason about programs (Murali et al., 2018; Allamanis et al., 2018) or limits the output to be within the scope of domain-specific languages (Lin et al., 2018; Yu et al., 2018; Desai et al., 2016).

In this work, we define a new task for autocompletion and present a simple, language-agnostic approach. A natural next step would be to incorporate contextual information into the synthesis process, so that the system can achieve comparable performance to the existing tools. Ultimately, we would like to customize the output of the system for each user based on local data, and be able to both autocomplete and search over source code. We hope that our work motivates the development of richer models and methods that can tackle this task.

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

## A. Examples of Successful Cases

| Input (keywords in parentheses are optional) | Desired and Synthesized Output |
|---|---|
| `for` | `for i in range():` |
| `for k` | `for k in range():` |
| `for 10` | `for i in range(10):` |
| `for l mylist` | `for l in mylist:` |
| `for l sorted mylist` | `for l in sorted(mylist):` |
| `for i len mylist` | `for i in range(len(mylist)):` |
| `for enumerate` | `for i, value in enumerate():` |
| `def main` | `def main():` |
| `def example` | `def example():` |
| `def my function` | `def my_function():` |
| `def f a b c` | `def f(a, b, c):` |
| `def f a="default"` | `def f(a="default"):` |
| `(def) init` | `def __init__():` |
| `(def) init self` | `def __init__(self):` |
| `(def) init kwargs` | `def __init__(self, **kwargs):` |
| `(def) init args kwargs` | `def __init__(self, *args, **kwargs):` |
| `class example` | `class Example():` |
| `class my exception` | `class MyException(Exception):` |
| `if x none` | `if x is None:` |
| `if x not none` | `if x is not None:` |
| `(if) isfile` | `if os.path.isfile():` |
| `(if) path not exists` | `if not os.path.exists():` |
| `(if) isinstance` | `if isinstance():` |
| `(if) isinstance mylist list` | `if isinstance(mylist, list):` |
| `(if) not isinstance mylist str` | `if not isinstance(mylist, str):` |
| `os` | `import os` |
| `sys` | `import sys` |
| `logging` | `import logging` |
| `argparse` | `import argparse` |
| `unittest` | `import unittest` |
| `numpy` | `import numpy as np` |
| `tensorflow` | `import tensorflow as tf` |
| `unittest test case` | `from unittest import TestCase` |
| `future division` | `from __future__ import division` |
| `future print` | `from __future__ import print_function` |
| `future absolute import` | `from __future__ import absolute_import` |
| `future unicode literals` | `from __future__ import unicode_literals` |
| `absolute division print` | `from __future__ import absolute_import` `division, print_function` |
| `except` | `except:` |
| `except io` | `except IOError:` |
| `except value` | `except ValueError:` |
| `except import` | `except ImportError:` |
| `except eof` | `except EOFError:` |
| `except attribute` | `except AttributeError:` |
| `except attribute a` | `except AttributeError as a:` |
| `except e` | `except Exception as e:` |
| `raise assertion` | `raise AssertionError()` |
| `raise implemented` | `raise NotImplementedError()` |
| `exit` | `sys.exit()` |
| `exit 1` | `sys.exit(1)` |
| `exit 2` | `sys.exit(2)` |
| `parser argparse` | `parser = argparse.ArgumentParser()` |
| `args parse` | `args = parser.parse_args()` |
| `add argument` | `parser.add_argument()` |

*Table A.* Examples of inputs and outputs (both desired and synthesized by the system) from the manually generated test set

| | |
|---|---|
| try | try: |
| finally | finally: |
| else | else: |
| pass | pass |
| while true | while True: |
| none | return None |
| true | return True |
| false | return False |
| assert true | assert True |
| assert false | assert False |
| i 256 | i = 256 |
| s "hello" | s = "hello" |
| if s "hello" | if s == "hello": |
| for s "hello" | for s in "hello": |
| print "hello" | print("hello") |
| self x | self.x = x |
| name = kwargs get | self.name = kwargs.get() |
| name = kwargs get "name" | self.name = kwargs.get("name", None) |
| f write data | f.write(data) |
| data f read | data = f.read() |
| data f read split | data = f.read().split() |
| open filename "r" | with open(filename, "r") as f: |
| mylist sort | mylist.sort() |
| newlist sorted mylist | newlist = sorted(mylist) |
| myline = " " join mylist | myline = " ".join(mylist) |
| newlist sorted list set mylist | newlist = sorted(list(set(mylist))) |
| super init | super().__init__() |
| (if) name "__main__" | if __name__ == "__main__": |
| self assert in | self.assertIn() |
| self assert not in | self.assertNotIn() |
| self assert raises | self.assertRaises() |
| self assert not equal | self.assertNotEqual() |
| self assert not none | self.assertIsNotNone() |
| self assert instance | self.assertIsInstance() |
| with self assert raises type | with self.assertRaises(TypeError): |
| logger get | logger = logging.getLogger() |
| logger (set) level | logger.setLevel() |
| logger debug "mymessage" | logger.debug("mymessage") |
| myarray 1 2 3 | myarray = [1, 2, 3] |
| myarray np 1 2 3 | myarray = np.array([1, 2, 3]) |
| with tf sess | with tf.Session() as sess: |
| with tf graph default | with tf.Graph().as_default(): |

*Table A.* [continue] Examples of inputs and outputs (both desired and synthesized by the system) from the manually generated test set

Table A shows 86 out of 143 (60.14%) input and output pairs that are successfully synthesized by our system. Note that the system can distinguish similar, but different input keywords, e.g., s "hello" and if s "hello", and be able to generate desired outputs s = "hello" and if s == "hello":, respectively.

## B. Examples of Erroneous Cases

Table B shows 26 out of 57 (45.61%) input and output pairs that our system could not synthesize as the top 1 suggestion. We include short description of the major errors in the examples and highlight relevant tokens. Specifically, tokens that are specified as input keywords, but are not included in the output, or tokens that are not desired, but inserted as a part of the output, are colored in red. Note that one example may contain multiple errors.

| Input | Desired and Synthesized Output (first and second line, respectively) |
|---|---|
| *Limited by common usage of code spanning more than one line* | |
| `parser add` | `parser.add_argument()` |
| | `parser.add_argument(` |
| *Fail to copy all input keywords* | |
| `for a b c tuples` | `for a, b, c in tuples:` |
| | `for a, b in tuples:` |
| `def f a=none` | `def f(a=None):` |
| | `def f(self, A, A):` |
| `def f self args kwargs` | `def f(self, *args, **kwargs)` |
| | `def f(self, args, *args):` |
| *Fail to format syntax* | |
| `self assert all equal` | `self.assertAllEqual()` |
| | `self.assert_all_equal()` |
| `myarray 1 2 3 4 5` | `myarray = [1, 2, 3, 4, 5]` |
| | `myarray = [[1, 2, 3], [4, 5]]` |
| `rst = filter none tmp` | `rst = filter(None, tmp)` |
| | `rst.filter(filter, tmp)` |
| `s = sorted mylist lambda x x[1]` | `s = sorted(mylist, key=lambda x:  x[1])` |
| | `s = sorted(mylist_mylist, x[x:x], x)[1]` |
| `s = sorted mylist lambda k v v k` | `s = sorted(mylist, key=lambda (k, v):  (v, k))` |
| | `s = sorted(mylist_mylist, lambda *k, **k:None)` |
| `codecs open filename "r" "utf-8"` | `with codecs.open(filename, "r", "utf-8") as f:` |
| | `with codecs.open(filename, "r") as "utf-8":` |
| *Insert additional tokens* | |
| `self assert true` | `self.assertTrue()` |
| | `self.assertIsTrue()` |
| `self assert false` | `self.assertFalse()` |
| | `self.assertIsFalse()` |
| `self assert equal` | `self.assertEqual()` |
| | `self.assertNotEqual()` |
| `self assert none` | `self.assertIsNone()` |
| | `self.assertIsNotNone()` |
| `class my class` | `class MyClass():` |
| | `class MyClass(Class):` |
| *Fail to insert proper tokens* | |
| `import tf` | `import tensorflow as tf` |
| | `import tf` |
| `import groupby` | `from itertools import groupby` |
| | `import groupby` |
| `from import groupby` | `from itertools import groupby` |
| | `from .  import groupby` |
| `from import copytree` | `from shutil import copytree` |
| | `from .  import copytree` |
| `import tensorflow layers` | `import tensorflow.contrib.layers as layers` |
| | `import tensorflow.layers` |
| `import tensorflow slim` | `import tensorflow.contrib.slim as slim` |
| | `import tensorflow as slim` |
| `import tensorflow lookup ops` | `from tensorflow.python.ops import lookup_ops` |
| | `import tensorflow.lookup.test_ops` |
| `import test case` | `from unittest import TestCase` |
| | `from test import TestCase` |
| `mylist sort reverse` | `mylist.sort(reverse=True)` |
| | `mylist.sort(reverse())` |
| `pickle dump data "save.p"` | `pickle.dump(data, open("save.p", "w"))` |
| | `pickle.dump(data="save.p")` |
| `data = pickle load "save.p"` | `data = pickle.load(open("save.p", "r"))` |
| | `data = pickle.load("save.p")` |

*Table B*. Erroneous examples from the manually generated test set. Tokens that are specified as input keywords, but are not included in the output, or tokens that are not desired, but inserted as a part of the output, are colored in red.