# OpenReview forum: "Autocompletion of Code from Keywords"
_ICML.cc/2018/Workshop/NAMPI — Submitted to NAMPI 2018_

### Review · AnonReviewer1 · 2018-06-14
**seq2seq to translate keyword sequence into a line of code**

**Rating:** 4
**Confidence:** 3

**Review:**

The authors propose to use a standard seq2seq pipeline to expand a sequence of keywords into a line of python code. The central contribution is a method to create a training dataset for this method from existing source code repositories.
Several methods to create this dataset are discussed; they all follow the "drop some tokens" pattern. The dropout probability is either uniform and random or proportional to the frequency of the keyword in the raw code dataset (intuitively, this means that common things get dropped more often). An evaluation on data generated like this, as well as on a small set of keyword/code samples generated manually by the authors, is presented.

Overall, I feel that the paper is not a good fit for the NAMPI workshop for three reasons:
(1) The workshop is focused on neural programming/program induction; whereas this paper is a "apply machine learning to program code" kind of work.
(2) The evaluation in this work is very unsatisfying,  as the manually generated tasks are often extremely unnatural ("true" -> "return True", "else" -> "else:", "f write data" -> "f.write(data)", etc). I'm not sure the results on this data are indicative of the overall usefulness of the approach
(3) The writing seems to quite unpolished. For example, the problem definition and the data generation do not match: Whereas Sect. 2 states that the keywords are meant to be a subset (and explicitly says that they do not need to be a subsequence), the data generation procedure only generates subsequences.

Finally, the authors should consider a comparison with highly related work such as https://arxiv.org/abs/1703.05698 and the very recent https://arxiv.org/abs/1805.04793.

---

### Review · AnonReviewer3 · 2018-06-25
**A clever take on code completion**

**Rating:** 7
**Confidence:** 3

**Review:**

Authors propose a novel take on autocomplete. Rather than the standard, predict-next-keyword problem, they train a seq2seq model (with attention and a copying mechanism) to reconstruct the original line of code after dropping out some of its keywords, in a Denoising Auto-Encoder fashion.

---

### Decision · ~NAMPI_Admin1 · 2018-06-28
**Paper5 Final Decision**

Reject